

# PhySortR: a fast, flexible tool for sorting phylogenetic trees in R

Timothy G. Stephens[1], Debashish Bhattacharya[2], Mark A. Ragan[1] and Cheong Xin Chan[1]

[1] ARC Centre of Excellence in Bioinformatics, and Institute for Molecular Bioscience, University of Queensland, Brisbane, Queensland, Australia

[2] Department of Ecology, Evolution and Natural Resources, Rutgers University, New Brunswick, NJ, USA

## ABSTRACT

A frequent bottleneck in interpreting phylogenomic output is the need to screen often thousands of trees for features of interest, particularly robust clades of specific taxa, as evidence of monophyletic relationship and/or reticulated evolution. Here we present PhySortR, a fast, flexible R package for classifying phylogenetic trees. Unlike existing utilities, PhySortR allows for identification of both exclusive and non-exclusive clades uniting the target taxa based on tip labels (i.e., leaves) on a tree, with customisable options to assess clades within the context of the whole tree. Using simulated and empirical datasets, we demonstrate the potential and scalability of PhySortR in analysis of thousands of phylogenetic trees without a priori assumption of tree-rooting, and in yielding readily interpretable trees that unambiguously satisfy the query. PhySortR is a command-line tool that is freely available and easily automatable.

# INTRODUCTION

Phylogenomics increasingly involves the screening of thousands of phylogenetic trees using specialised sorting algorithms that assign phylogenetic trees a classification based on features of interest, e.g., strongly supported monophyletic relationships of taxa in question (i.e., the "target" taxa). Here, phylogenetic trees in flat files (e.g., Newick format) are sorted (i.e., classified) based on text-pattern matching. This principle is not to be confused with the *tree sort* process, common in computer science, of rearranging binary data elements in an ordered structure (*Knuth, 1971*). Currently available utilities, e.g., PhyloSort (*Moustafa & Bhattacharya, 2008*) and SICLE (*DeBlasio & Wisecaver, 2013*) screen a set of phylogenetic trees for the presence of clades that unite a set of user-defined target taxa (as indicated in tip labels, i.e., *leaves*, on the tree) based on clade support that exceeds a defined threshold, and sort these trees accordingly; SICLE (*DeBlasio & Wisecaver, 2013*) specifically identifies all nearest neighbours (sister clades) of a single user-defined target. However, these tools do not consider the proportion of non-target leaves and overall taxon composition in a tree during the sorting process. Moreover, tools implemented in a graphical user interface e.g., PhyloSort (*Moustafa & Bhattacharya, 2008*) do not allow for automation of multiple analyses, thus limiting scalability.

Corresponding author
Cheong Xin Chan,
c.chan1@uq.edu.au

Here we present PhySortR, a fast, flexible R package for screening and sorting phylogenetic trees. The command-line package provides the quick and highly flexible *sortTrees* function, allowing for screening (within a tree) for "Exclusive" clades that contain only the target leaves and/or "Non-Exclusive" clades that include a defined portion of non-target leaves. Using simulated data, we assess the runtime of PhySortR based on the number of trees and the number of leaves within a tree, and demonstrate the potential of PhySortR in the analysis of multiple, large-scale empirical datasets.

## MATERIALS & METHODS

### Rationale and basic principles of PhySortR

Figure 1 shows four examples of tree topologies and their corresponding features relevant to the sorting process of PhySortR, each with a target clade identified as Clade *Z*; the first tree (Fig. 1A) is an empirical tree of a putative sodium/sulphate symporter protein from an earlier study (*Bhattacharya et al., 2013*), on which the other three hypothetical topologies (Figs. 1B–1D) are based. PhySortR allows the user to specify one or more target terms using the *target.groups* argument, providing that the leaves (tree-tip labels in the Newick files) are named consistently across all input trees; this is a simple string-matching exercise, i.e., the terms specified here determine taxon-level resolution of targets. In the examples shown in Fig. 1, *target.groups* = "Rhodophyta,Viridiplantae,Stramenopiles." The minimum support for a clade (*min.support*) can refer to bootstrap, Bayesian posterior probability, or any other measure of support. Here, no prior assumption of tree root is made (i.e., all trees are treated as unrooted), thus clade membership on both sides of each node is considered. For instance, the node support for Clade *Z* (bootstrap 99%) in Fig. 1A is also considered as the support for the opposing bacterial clade (Proteobacteria + Cyanobacteria) on the tree. This is distinct from the assumption of the lowest common ancestor for a clade in PhyloSort (*Moustafa & Bhattacharya, 2008*), in which subtree-rooting could be invoked during a search.

Existing utilities identify clades of interest without considering the occurrence of the target leaves elsewhere on the tree. For example, when assessing a target clade in a 30-leaf tree, PhyloSort will positively identify both (a) a robust 24-member clade (Clade *Z* in Figs. 1A) and (b) a robust four-member clade (Clade *Z* in Fig. 1B), although (a) is the more-convincing evidence of a close association between the targets and of greater biological significance; here most of the target group is contained within Clade *Z* (Fig. 1A), compared to the scenario in Fig. 1B, in which most of the target group, with no clear evidence of overall monophyly, is placed externally to robust Clade *Z*. To address this issue, PhySortR allows the user to define *min.prop.target*, the minimum required proportion of target(s) present in a clade relative to the total number of target(s) found in a tree. The default value of *min.prop.target* is set at 0.7 as guidance. At 1.0, a strict monophyletic relationship of the target group is enforced (i.e., no target leaves occurring elsewhere on a tree). When the value is set too low (e.g., at 0.1), one would identify a target clade that is more-narrowly defined (e.g., Clade *Z* in Fig. 1B); such a clade could have limited biological significance. This option provides users the flexibility to design a query that can be tailored to address

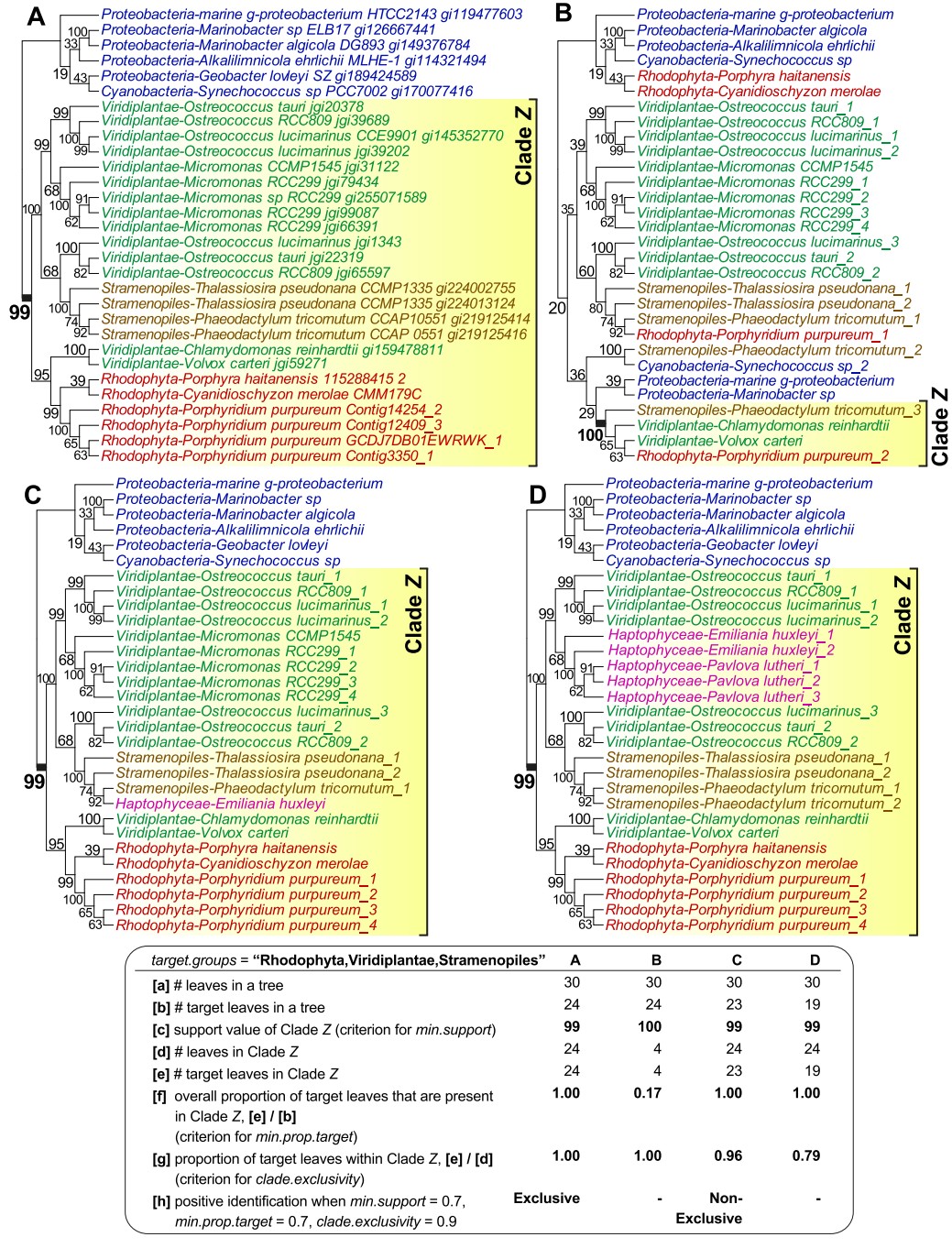

**Figure 1** **The effect of minimum target proportion and clade exclusivity on a tree.** Tree topologies and their corresponding features relevant to *min.prop.target* and *clade.exclusivity* in PhySortR, with a target clade *Z*. The tree of a sodium/sulphate symporter protein (A) contains a strongly supported "Exclusive" clade *Z* of Rhodophyta + Viridiplantae + Stramenopiles. Different scenarios are shown using hypothetical tree topologies, for (B) a low target proportion within target clade, (C) a "Non-Exclusive" target clade with a high extent of clade exclusivity, and (D) a low extent of clade exclusivity. A summary of the leaf composition corresponding to key selection criteria of PhySortR for each tree is shown at the bottom.

specific biological questions, e.g., to identify a clade of interest defined as narrowly or as broadly as desired.

Alternatively, the user may wish to screen for a robust clade that contains the target groups and a small proportion of "interrupting" non-target leaves, e.g., a 24-member clade consisting of 23 leaves from the targets Rhodophyta, Viridiplantae, and Stramenopiles, as well as one from a non-target leaf, Haptophyceae (Clade *Z* in Fig. 1C). Whereas the haptophyte is "interrupting" the otherwise exclusive clade of Rhodophyta + Viridiplantae + Stramenopiles, the association among the targets is still of interest and the presence of the haptophyte might be readily explained by lateral gene transfer (LGT) due to plastid endosymbiosis (e.g., *Bowler et al., 2008*; *Chan et al., 2011b*; *Howe et al., 2008*). Composite clades such as these are considered "Non-Exclusive" (*Chan et al., 2011b*) and are not identified by existing sorting tools. The concept of exclusivity (Fig. 1A) versus non-exclusivity (Fig. 1C) of clades in tree sorting has proven crucial in a number of genome-wide studies that have investigated the impact of LGT on the evolution of diverse algae and protists (e.g., *Bhattacharya et al., 2013*; *Chan et al., 2011b*; *Curtis et al., 2012*; *Price et al., 2012*). In addition to biological implications (e.g., LGT or genetic recombination), by allowing the presence of non-targets in a clade, the non-exclusive clades are also useful in identifying the association of a target group against the backdrop of phylogenetic artefacts (*Stiller, 2011*) that would weaken an otherwise strong phylogenetic signal, e.g., unbalanced taxon sampling (or missing taxa) (*Rosenberg & Kumar, 2003*; *Sanderson, McMahon & Steel, 2010*), long-branch attraction (*Felsenstein, 1978*), or contamination. PhySortR identifies both types of clade based on the proportion of target versus non-target leaves using the option *clade.exclusivity*. At the default setting (*clade.exclusivity* = 0.9), the minimum proportion of target leaves within a "Non-Exclusive" clade is 0.9, thus the maximum proportion of non-target leaves allowed in the clade is 0.1 (i.e., 1 minus 0.9). For instance, the proportion of target leaves within Clade *Z* (0.79) in Fig. 1D does not satisfy the criterion of *clade.exclusivity* of 0.9; this clade is therefore not considered as "Non-Exclusive" at the default setting. This option accepts any value <1.0, and is applicable only for sorting "Non-Exclusive" clades (see below); at 1.0 (no non-target leaves allowed), the clade is considered "Exclusive".

## Sorting of phylogenetic trees

In PhySortR, *sortTrees* is the function for sorting phylogenetic trees; the basic algorithm is shown in Fig. 2, and all available arguments are detailed in Table 1. To run *sortTrees*, the user must aggregate all phylogenetic trees to be sorted into a single directory. All tree files must have an identical file extension (see *extension*; Table 1) and can be in either standard or extended Newick (*Cardona, Rossello & Valiente, 2008*) format.

The *target.groups* parameter (Table 1) is the only compulsory argument; all other arguments have defaults that the function will use if an alternative is not provided. To avoid ambiguity, the terms passed to the function are matched to a tree's tip labels by exact substring-matching. Multiple terms passed to the function must be separated by a comma (e.g., "*Rhodophyta,Viridiplantae,Stramenopiles*") and must be sufficiently specific in the dataset for the purpose of the screening. For instance, "*plantae*" and "*Viridiplantae*" might

1   **Input:** Phylogenetic trees and user-defined sorting parameters

2   **Output:** Sorted phylogenetic trees

3   **for** each *tree* in target directory **do**

4      read *tree*

5      **if** *tree* is in extended Newick format **then**

6         convert into traditional Newick using *convert.eNewick*

7      **end if**

8      import *tree* using *ape* package

9      **if** all taxa in *tree* are targets **then**

10        the *tree* is "All Exclusive"

11     **end if**

12     **for** each rooted and unrooted node in *tree* **do**

13        **if** node satisfies *min.support, min.prop.target* **and**
            node has ≥ 1 taxon from each member of *target.groups* **then**

14           **if** all target taxa in the node **then**

15              the *tree* is "Exclusive"

16           **else if** node satisfies clade.exclusivity **then**

17              the *tree* is "Non-Exclusive"

18           **end if**

19        **end if**

20     **end for**

21  **end for**

**Figure 2   Overview of the sorting algorithm in PhySortR.**

not be appropriate in a single query because all tips that are identified by "*Viridiplantae*" will also be identified by "*plantae*."

Regardless of which parameters are passed to the *mode* argument, the function will always return a list of the trees that have been identified as containing clades that meet the specified criteria. If the move (*mode* = "m") or copy (*mode* = "c") command is given, subdirectories will be created in *out.dir* that contain trees with a particular clade, i.e., the directory *out.dir/Exclusive/* will be created for the trees with "Exclusive" clades and *out.dir/Non_Exclusive/* for trees with "Non-Exclusive" clades. If the function is instructed to search for "Exclusive" trees it will also return trees that contain only target leaves, termed "All Exclusive" trees. These trees are a subset of "Exclusive" trees and will be transferred

**Table 1   List of arguments within _sortTrees_ function in PhySortR.**

| Argument | Description |
| --- | --- |
| target.groups | A set of one or more terms that represent the target leaves. Multiple terms are to be separated by a comma and enclosed in quotation marks i.e., "_Rhodophyta,Viridiplantae_". This process is case-sensitive; it uses partial string matching, so the terms used must be unique i.e., "_plantae_" and "_Viridiplantae_" are not appropriate as the first is a subset of the second. |
| min.support | The minimum support (between 0–1 or 0–100 inclusive) for any clade identified during sorting (default 0), dependent on the range of support values noted in the tree file (e.g., bootstrap support, Bayesian posterior probability, or any similar measure). A node with no identified support value is treated as having a value of zero (0). |
| min.prop.target | The minimum proportion (between 0.0–1.0 inclusive) of target leaves to be present in a clade, out of the total target leaves in the tree (default 0.7). At 0.5, $\geq$50% of all target leaves in a tree must be in the clade; at 1.0, all target leaves in a tree must be in the clade. |
| in.dir | The path to the input directory containing all phylogenetic trees to be sorted. If no value is given, the function defaults to the user's current working directory. |
| out.dir | The output directory to be created within _in.dir_, for the trees identified during sorting to be moved or copied to. If _out.dir_ is omitted, the default directory of _Sorted_ Trees/_ will be used. If list mode (mode = "_l_") is given, this argument will be ignored, and no directory will be created. The content of _out.dir_ is dependent on the _clades.sorted_ parameter. |
| mode | Option to control whether the function will move ("_m_"), copy ("_c_") or list ("_l_") the files containing trees identified during sorting. In both move and copy modes the files will be transferred to subdirectories within _out.dir_ and a list of the sorted trees will be returned. In the list mode, only the list will be returned. The type of trees sorted is dependent on the _clades.sorted_ parameter. |
| clades.sorted | Option to control sorting for "Exclusive" ("_E_") or "Non-Exclusive" ("_NE_") clades. The default setting is to search for both types of clades, i.e., "_E,NE_". Sorting of "Exclusive" clades will also generate a sub-group of "All Exclusive" trees. This argument will affect what is returned by the function and what subdirectories are created in _out.dir_. |
| extension | The file extension of the input phylogenetic trees (default "_.tre_"). |
| clade.exclusivity | The minimum proportion of target leaves allowed in a "Non-Exclusive" clade, applicable only when sorting _NE_ clades. The value must be $\geq$ 0.0 and <1.0. At default (0.9), $\geq$90% (but not 100%) of the leaves in a _NE_ clade must be target leaves (i.e., <10% can be "interrupting" non-target leaves). Specification of 1.0 is not allowed; 1.0 implies that all (100%) leaves in a clade are target leaves (no non-target leaves allowed), thus the clade would be "Exclusive," not "Non-Exclusive." |

Newick:
(A:0.1,(B:0.2,(C:0.3,D:0.4)100:0.5)95:0.55);
Extended Newick (eNewick):
(A:0.1,(B:0.2,(C:0.3,D:0.4)0.5[100])0.55[95]);

**Figure 3** **A phylogenetic tree represented in standard and extended Newick formats.**

to a subdirectory (if the move/copy parameter is given) within the "Exclusive" directory i.e., *out.dir/Exclusive/All_Exclusive*.

The *clades.sorted* parameter can be used to change the types of clades that the function will search for. For example if *clades.sorted* = "*E*" is given, the function will only search for trees that have "Exclusive" clades, but if the default value of *clades.sorted* = "*NE,E*" is given, the function will search for both "Exclusive" and "Non-Exclusive" clades. During each run the function will create a log file, called "*out.dir*.log," in the *in.dir* directory. This file will contain information about each identified clade, e.g., the names of the leaves in the clade, the support for the clade, the proportion of "interrupting" leaves, and so forth.

## Conversion of extended Newick format

Newick format is a standardised, machine-readable, plain-text representation of phylogenetic trees (http://evolution.genetics.washington.edu/phylip/newick_doc.html) that has been widely adopted in phylogenetic software. This format was later modified to incorporate more-complex network information such as hybrid nodes, in the form of extended Newick (eNewick) format; see *Cardona, Rossello & Valiente (2008)* for details. The two formats are however very similar, as shown in the example tree topology in Fig. 3. In Newick, the support value for a node precedes its branch length, separated by a colon. In eNewick, the support value for a node, enclosed in square brackets, is placed after its branch length (Fig. 3). Most phylogenetic programs accept trees only in the Newick format, but more-recent programs of phylogenetic inference generate the tree output in eNewick format by default. Taking this into consideration, PhySortR provides the *convert.eNewick* function that takes a single phylogenetic tree in eNewick format and returns the same tree in Newick format. This function in isolation can be used as a general-purpose tool for converting phylogenetic trees in eNewick format into a format that is usable by the popular phylogenetic packages in R, *ape* (*Paradis, Claude & Strimmer, 2004*) and *phytools* (*Revell, 2012*).

## Simulation of phylogenetic trees

To test the scalability of the PhySortR package we simulated benchmarking datasets composed of a given number of trees ($N$) and leaves per tree ($X$); see Fig. S1 for detail. All simulated trees are in the eNewick format. To simulate a tree with $X = 100$, we used

a base phylogenetic tree with $1.05X$ tips, i.e., 105 tips. An "Exclusive" 20-leaf target clade (i.e., $0.2X$) is defined, tip labels of other non-target leaves are swapped (at random), following which $0.05X$ (i.e., 5) of the overall tree branches (external to the target clade) chosen at random were removed using *phytools*, resulting in the final tree of size $X$. This tree was then replicated up to $N$ number of trees as per our experimental design below.

Simulation of trees at different $X$ follows the same strategy as per above, and for negative controls, the target clade was simply omitted. For the first analysis, we generated sets of input trees at $N = 1,000, 2,000, 4,000, 6,000, 8,000$ and $10,000$ (each tree with $X = 100$; Data S1). For the second analysis, we generated sets of input trees ($N = 1,000$) at tree size $X = 100, 200, 300, 400$ and $500$ (each tree of distinct size $X$ is available as Data S2). For the purpose of assessing scalability, all tree topologies are identical within a set of $N$ trees, i.e., a single technical replicate (Fig. S1). All benchmark analyses were carried out with 100 technical replicates (i.e., each replicate with a distinct base tree topology), on a desktop computer (2.5 GHz Intel® Core™ i5 with 4 GB memory). The R script for simulating these topologies and the corresponding templates are available as Data S3.

### Comparative assessment of PhySortR versus PhyloSort

As input, here we used 897 empirical protein trees from an earlier study (*Chan, Reyes-Prieto & Bhattacharya, 2011a*). Using PhySortR and PhyloSort (http://phylosort.sourceforge.net/), we screened for trees that contain a strongly supported (bootstrap $\geq 90\%$) exclusive clade of Viridiplantae + Stramenopiles. The closest equivalent parameter settings were used for PhySortR (*target.groups* = "Viridiplantae,Stramenopiles", *min.support* = 90, *min.prop.target* = 1.00, *clades.sorted* = "E") and PhyloSort (*Taxa regexp* = ([^_]+_[^_]+).*, *query taxa group 1* = all Viridiplantae leaves, *query taxa group 2* = all Stramenopiles leaves, *Min bootstrap* = 90, *Exclusive mode*). For PhyloSort, independent analysis was done with the option *Root by outgroup* (PhyloSort-root hereinafter) and without (PhyloSort-no-root).

### Implementation and availability of PhySortR

PhySortR depends on two other phylogenetic packages in R, *ape* (*Paradis, Claude & Strimmer, 2004*) and *phytools* (*Revell, 2012*). All three packages require R version 3.0 or above to function, and they can be installed directly from CRAN in the R environment (see Text S1 for detail). PhySortR is freely available as a platform-independent R package from the Comprehensive R Archive Network at https://cran.r-project.org/web/packages/PhySortR/. Several examples are provided with the R package.

## RESULTS & DISCUSSION

Figure 4 shows the runtime of PhySortR relative to the number of trees ($N$) to be sorted and the number of leaves ($X$) within a tree. As with any utility (*DeBlasio & Wisecaver, 2013*; *Moustafa & Bhattacharya, 2008*), the runtime of PhySortR is dependent on $N$ and $X$. We observed that the runtime scales linearly with $N$ (Fig. 4A) and superlinearly with $X$ (Fig. 4B). In the extreme case, sorting through 10,000 trees of $X = 100$ took <400 s ($\sim$6.7 min), and sorting through 1,000 trees of $X = 500$ took <350 s ($\sim$5.8 min). We

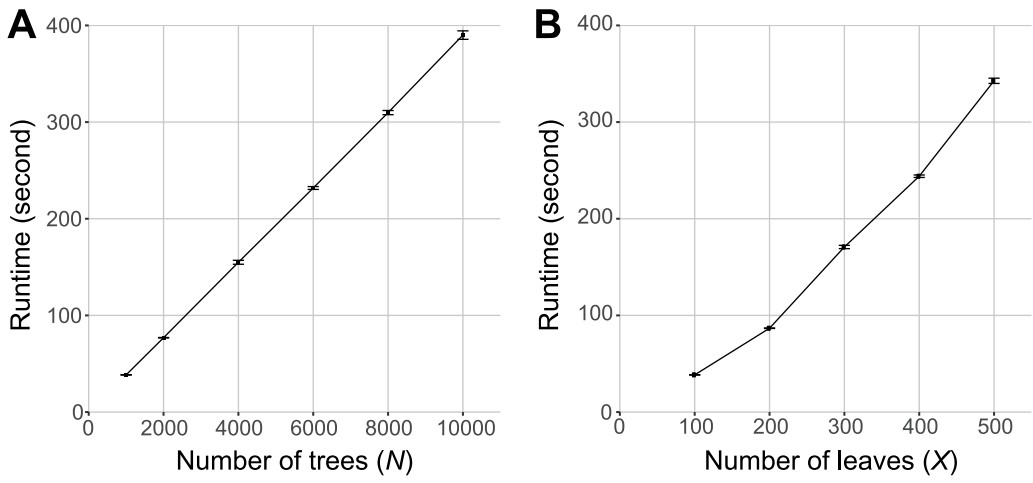

**Figure 4 Benchmarking results of PhySortR using simulated data.** The mean runtime of PhySortR is shown for analysis across datasets (A) with different numbers of trees, $N$, and (B) with different numbers of leaves per tree, $X$. Values of runtime (in second) are mean across 100 replicates, error bars indicate the standard deviation of the mean.

observed negligible differences in runtime with negative controls (trees containing no identifiable clades) as input, compared to the test set in Fig. 4B. Our findings demonstrate the potential of PhySortR in analysis of multiple, large-scale datasets.

Unlike PhyloSort, no prior assumption of tree-rooting is made in PhySortR. To illustrate the impact of tree-rooting assumption on the sorting process, we performed sorting among 897 empirical protein trees (*Chan, Reyes-Prieto & Bhattacharya, 2011a*) using PhySortR, and compared them to PhyloSort with and without the rooting option (i.e., PhyloSort-root and PhyloSort-no-root; see 'Materials & Methods' for detail). Here, in the search for an "Exclusive" clade of Viridiplantae + Stramenopiles with bootstrap ≥90%, 18, 33 and 46 trees were identified using PhyloSort-no-root, PhySortR and PhyloSort-root (Fig. 5A). In Fig. 5B, identification of the target clade (bootstrap 98%) is straightforward in all methods; all methods also successfully recovered the same ten "All Exclusive" trees (containing only leaves of Viridiplantae + Stramenopiles). Figure 5C shows a tree that is identified only using PhyloSort-root. Here the target clade is positively identified due to (the enabled) subtree-rooting of non-target leaves, not based on clade support. The absence of other closely related leaves in the tree could reinforce the association of Viridiplantae and Stramenopiles (i.e., the tree might be biologically meaningful), but the bootstrap 61% (Fig. 5C) is below the specified threshold (90%) in the search; thus this tree is not recovered using PhySortR and PhyloSort-no-root. Figure 5D shows a tree that is identified using PhySortR and PhyloSort-root, but not PhyloSort-no-root. Here the condition of target clade support is satisfied (i.e., bootstrap 98%; Fig. 5D), but with the subtree-rooting option disabled, PhyloSort did not recover this tree in the search; this appears to be a false negative. Whereas the results from PhyloSort and PhySortR are not directly comparable, our results demonstrate the impact of the tree-rooting assumption on the sorting results between the two programs. This underlying assumption in PhyloSort could lead to under- or over-estimation of the number of positive identifications; instances of false positives
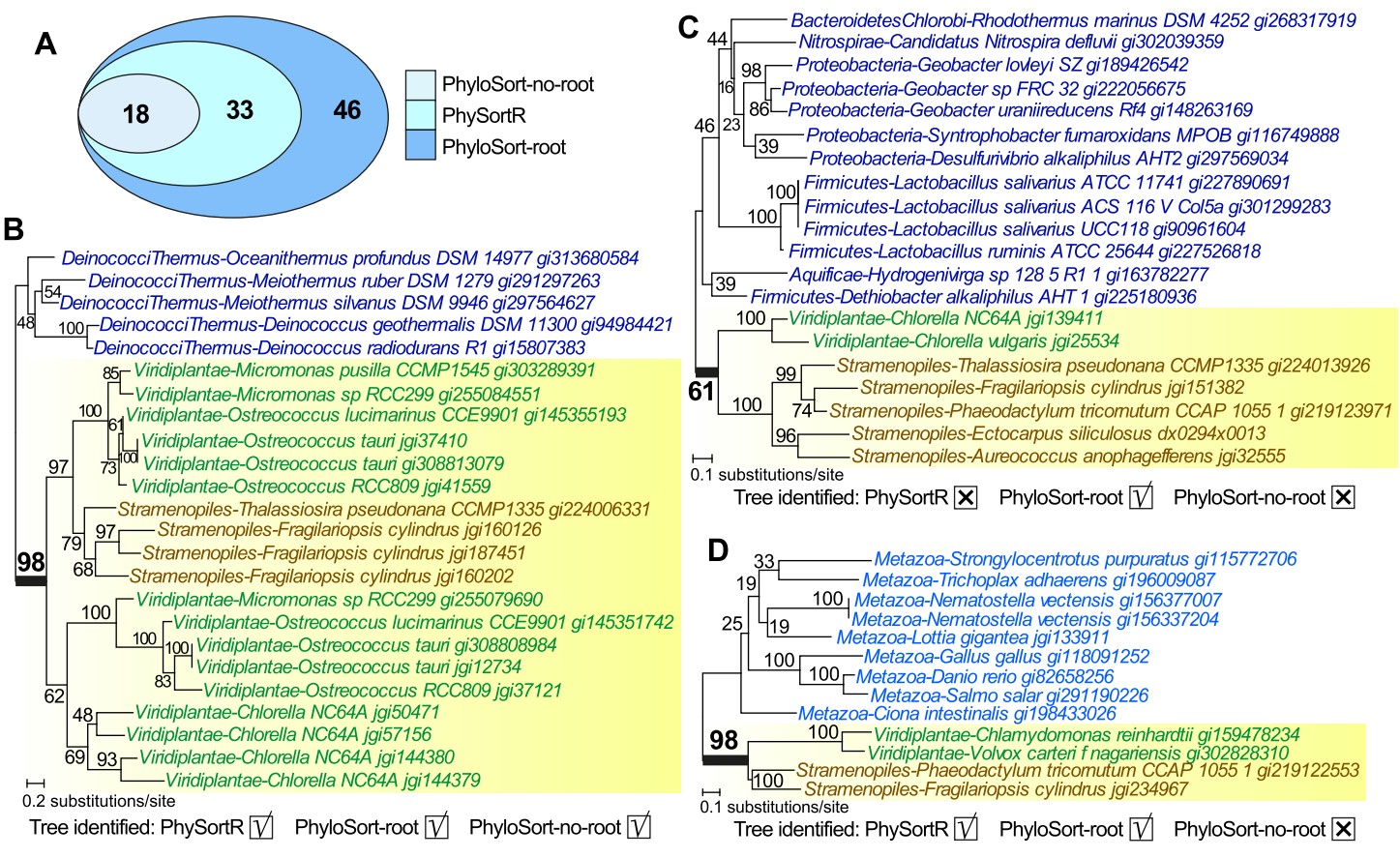

**Figure 5 Trees sorted using PhySortR and PhyloSort.** The Venn diagram depicting number of trees identified using PhySortR, PhyloSort-root and PhyloSort-no-root is shown in (A). An example of tree output for trees that are identified in (B) all cases, (C) only PhyloSort-root, and (D) all but PhyloSort-no-root are shown.

and false negatives will need to be manually verified based on the research question. Here, PhySortR yields readily interpretable trees (e.g., Figs. 5B and 5D) that unambiguously satisfy the query requirements.

Furthermore, the algorithm of PhySortR (Fig. 2) is distinct from existing utilities in two key aspects: PhySortR considers in a tree (a) both exclusive and non-exclusive clades, and (b) a clade of interest within the context of overall taxon composition. Both PhySortR and PhyloSort were designed as tools for hypothesis testing, e.g., to identify a clade of target leaves as putative evidence of genetic exchange and/or transfer. In comparison, SICLE was designed for a fundamentally different task, the screening of all possible sister clades to a single target group (instead of two or more targets as allowed in PhySortR and PhyloSort) as a tool for hypothesis generation. These three programs are implemented in different programming languages, i.e., PhySortR in R, PhyloSort in Java and SICLE in C++. The sorting process of phylogenetic trees is therefore dependent not only on parameter settings but also implementation and hardware; comparing computation time and results among these tools is not straightforward.

PhySortR is an R implementation based on the basic sorting principles of *Chan et al. (2011b)* that has been widely adopted in other phylogenomic studies (*Bhattacharya et al., 2013*; *Curtis et al., 2012*; *Price et al., 2012*). PhySortR incorporates existing functionalities and data structures in the commonly used phylogenetic packages *ape* (*Paradis, Claude & Strimmer, 2004*) and *phytools* (*Revell, 2012*), allowing for streamlined interoperability within the R environment. Whereas *ape* and *phytools* accept only Newick as input, PhySortR accepts tree files in both Newick and eNewick (*Cardona, Rossello & Valiente, 2008*) formats. The R platform (*R Core Team, 2015*) is open source, platform-independent, and broadly accessible to researchers, with continued support. In addition, functional modularity and the command-line interface of PhySortR enable batch automation and workflow integration.

## ACKNOWLEDGEMENTS

We thank two anonymous reviewers for their constructive comments and suggestions.

### Funding

This work was supported by the Australian Research Council Discovery Project (DP150101875) grant awarded to MAR, CXC and DB. TGS is supported by an Australian Postgraduate Award. CXC is supported by a Great Barrier Reef Foundation Bioinformatics Fellowship awarded to MAR. DB acknowledges support from the National Science Foundation (1004213). The funders had no role in study design, data collection and analysis, decision to publish, or preparation of the manuscript.

### Grant Disclosures

The following grant information was disclosed by the authors:
Australian Research Council Discovery Project: DP150101875.
Australian Postgraduate Award.
Great Barrier Reef Foundation Bioinformatics Fellowship.
National Science Foundation: 1004213.

### Competing Interests

The authors declare there are no competing interests.

### Author Contributions

- Timothy G. Stephens and Cheong Xin Chan conceived and designed the experiments, performed the experiments, analyzed the data, contributed reagents/materials/analysis tools, wrote the paper, prepared figures and/or tables, reviewed drafts of the paper.
- Debashish Bhattacharya and Mark A. Ragan contributed reagents/materials/analysis tools, reviewed drafts of the paper.

### Data Availability

Source code and manual of this toolkit are freely available from Comprehensive R Archive Network (CRAN): https://cran.r-project.org/web/packages/PhySortR/.

## Supplemental Information

Supplemental information for this article can be found online at http://dx.doi.org/10.7717/peerj.2038#supplemental-information.

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
