# Peer review of "PhySortR: a fast, flexible tool for sorting phylogenetic trees in R"

_PeerJ, doi:10.7717/peerj.2038_

## Round 0.1 · original submission · Major Revisions

Both reviewers felt that the manuscript provides a useful tool, but substantial additional work needs to be done to show that it works as intended. A comparison of how it performs relative to other methods in terms of running time would also be useful.

Personally, I would like to see a repository with test data and code, which can be used to evaluate the package's performance and the authors claims, and help reviewers test the package. It is crucial to show that solutions obtained using PhySortR match those obtained using other packages.

The reviewers also made a number of smaller suggestions, which I feel would greatly improve the manuscript, and potentially even increase the user base of this software.

Reviewer 1 ·

Basic reporting

In this software, they use the term “Sort” to mean “Classification” if I understand correctly, they attempt to stay consistent with other tools (i.e. PhyloSort) but as a Computer Scientist this means something else. A brief explanation of this in the abstract or intro might make the paper more accessible to the CS community.

The basics setup of PeerJ states that the acknowledgements section “Should not be used to acknowledge funders – that information will appear in a separate Funding Statement on the published paper.”

A brief explain of extended newick format (line 99) would move closer to being a self contained piece of work.

Experimental design

In table 1, and paragraph at line 79. the explanation of “target.groups” is somewhat confusing. If you cannot include “Taxon1” and “Taxon10” because one is a prefix of the other, than shouldn’t you also not include “Taxon1” and “Taxon2”? Also does the author mean to say prefix rather than substring? Finally, are the authors using regular expression matching or simple string matching (can the user input regular expressions)?

The explanation of how the replicate trees created during simulation starting at line 108 was hard to follow. A visual may help. Did the authors guarantee the same tree was not generated multiple times (the same 5 taxa removed)? The source code for the script used may be a nice addition.

Validity of the findings

This study is very interesting, and provides another tool for high-throughput phylogenetic analysis. I feel the study presented here is slightly inadequate for publication by itself. Seeing running time alone is interesting but does not really help a reader understand wall time in comparison to other methods. I understand that the time for PhyloSort are infeasible but you should be able to get the timings from SICLE to compare with. You could also estimate the time to compute the classification of one tree using PhyloSort and extrapolate to give an idea of the standard running time of that program.

It would be nice to provide confirmation that the software released is able to confirm results found by other pieces of software (i.e. of the trees produced, making sure that all of them are “Exclusive” in PhySortR, PhyloSort and SICLE). You are presenting a new realization of an existing algorithm, but the reader needs to know that the implementation is consistent with previous versions.

Additional comments

Figure 1 could be more clear, maybe using background shading or coloring to differentiate leaves with X and Y

Reviewer 2 ·

Basic reporting

See #1 and #2 in Comments for the Author

Experimental design

See #3, #4 and #5 in Comments for the Author

Validity of the findings

No Comments

Additional comments

The manuscript by Stephens et al presents the R package PhySortR, a tool for searching through a set of phylogenies and identifying the set of trees that contain a phylogenetic association of interest. The command-line nature of the tool and its ability to detect non-exclusive associations mitigates some of the methodological and biological challenges that plague high-throughput phylogenetics and merits its publication. However, there are several places where I feel the manuscript could use additional clarification.

Major comments
1. Allowing for the user defined association to be 'non-exclusive' is the real selling point of PhySortR. However the authors only highlight one biological process (endosymbiotic gene transfer) when justifying this feature's necessity. In fact, there are many other aspects of high-throughput phylogenetics, both biological processes and methodological issues, that make this a very nice feature (e.g. other forms of LGT, sequencing contamination, taxon sampling, etc). The authors should devote a section of the introduction to these challenges as context and justification for their program.

2. In the introduction, the authors state that all currently available tree-sorting utilities screen a set of trees for a user-defined association of interest (e.g. Identify all trees in which 'Viridiplantae' and 'Rhodophyta' are monophyletic. Here, the 'target' is the association being evaluated). However, the program SICLE, which the authors cite, actually describes all sister clades to a node of interest and does not take a user-defined association as input. For example, in SICLE, the target could be 'Viridiplantae' and the program would return all the nearest neighbors to 'Viridiplantae' across all input trees. 'Rhodophyta' would likely be a neighbor in many trees, but the program would return other associations as well. The difference may be subtle, but PhyloSort and PhySortR are programs for hypothesis testing, whereas SICLE is for hypothesis generation and bulk summary. Both approaches are valid and useful, but the distinction should be stated in the paper.

3a. Lines 47-54 starting with "Existing utilities..." to the end of the paragraph are very confusing. The hypothetical 100-taxon tree is not described in enough detail to make the example helpful. Is the tree made up entirely of members from the two target groups? Are sequences from both targets interleaved, or is each target monophyletic? An example figure would be helpful here (see comment #4), as would a real world example such as the one presented in the second methods paragraph (lines 55-60).

3b. The min.prop.target argument allows PhySortR to identify 'clades' of interest even if the target of interest does not form a monophyletic group in the tree (even when clades.sorted is set to exclusive). I realize the phylogenetic challenges mentioned above may necessitate such an argument be included, but the authors should explain and justify this more. My concern being, if min.prop.target is set too low, then interpreting the biological significance of any returned clade becomes very challenging.

4. Figure 1 should be expanded to include the full phylogenies, and the authors should demonstrate clade.exclusivity as well as min.prop.target arguments in these diagrams. As it is now, the reader doesn't know if taxa X and Y are present elsewhere in the phylogeny and to what extent.

5. The authors refer to the terminal nodes of trees as 'taxa', but this is only appropriate if the trees being evaluated represent species phylogenies. The input for PhySortR could be gene trees in which a single taxon may be represented by multiple sequences in an alignment and hence multiple nodes. This is particularly important for clarity when discussing the benchmarking results presented in Figure3. Therefore, the authors should use a less specific term in place of taxa/taxon (e.g. leaves).

Minor Comment
6. It seems like the program would be more intuitive if the clades.sorted and clade.exclusivity were collapsed into one argument. If this was implemented, when clade.exclusivity is set to 1.0 the clades are exclusive.

---

## Round 0.2 · Minor Revisions

Both reviewers found the manuscript to be greatly improved upon revision, though several minor issues remain. I am confident the authors can address them as well.

Reviewer 1 ·

Basic reporting

The authors address all of the concerns brought up in the original review of this paper. I have noted some items that were not quite addressed sufficiently.

Experimental design

In response to my comments about the leaf label searching method used, I think the additions made are reasonable. But just a minor note that "strict string-matching" is not quite accurate. I think you mean exact string matching (even more correct might be exact substring matching).

In response to my comment about the simulation data, the authors make the scripts to generate the tree topologies, they have made them available to the reviewers and mention that it is available upon request. While I agree that the information is variable so it may change over time as to how the users generate the data, this paper should stand on its own as a statement of how the experiment was performed at this time. Therefore the scripts should be posted as is as a record of this time point. In addition it should be kept as a record of the experiments performed so they can be independently verified by future readers.

Also Figure S1 is and section about simulating the data, it is still unclear why the trees are duplicated (I assume exactly) N times.

On line 103, you state that the exclusivity value should be <1. (1) I would state that it should be in the range [0,1) and (2) mention the value cannot be 1 (its actually programatically excluded). This is in Table 1, but I think it should be explicit in text if you're going to bring it up. As an aside I feel like this should be allowed and get rid of the "exclusivity" input parameter, but this is just personal preference.

Validity of the findings

I disagree with the fact that because different programs are in different languages, they cannot be compared. This is simply not true. While I agree that PhySortR, PhyloSort and SICLE are created for different purposes they can be compared. For instance, SICLE will tell you if a search criteria is monophyletic (i.e. search for an exclusive subtree with a certain prefix) and will print "CLADE NOT FOUND".

Additional comments

Line 118 has an unmatched parenthesis.

Reviewer 2 ·

Basic reporting

The authors seem to have responded well to the majority of the referee suggestions/comments. However, I do have some minor suggestions for improving the quality of the manuscript further.

1. Several of my comments as well as those of the editor and other reviewer were taken verbatim and placed within the revised manuscript. I can't speak for the others, but my language was generally casual and not something I intended for publication (e.g., lines 77-78, line 224). Along those lines, I think the manuscript would benefit from additional proofreading to improve its flow and diction.

2. Line 77; If the authors' acknowledge that a low min.prop.target (e.g. 0.1) would have low biological significance, when why allow the parameter to be set that low in the first place? I'm playing the devil's advocate here, because I think the tool is obviously more generalizable if the authors' do not set an arbitrary threshold. But I think the topic deserves additional clarification and rationalization here if the authors' expect their tool to be applicable to a general audience.

3. I appreciate the authors switching to leaves instead of taxa in many places, however the word taxa is still used in parts of the manuscript such as in figure 1 (where there are clear paralogs and therefore the leaves are not representative of taxa) and in figure 4.

Experimental design

no comments

Validity of the findings

no comments

---

## Round 0.3 · accepted · Accept

I believe that the authors made all the necessary changes for the manuscript to be published.

Reviewer 1 ·

Basic reporting

The authors have adequately responded to all of my concerns.

Experimental design

No comments

Validity of the findings

No comments

Additional comments

No comments